# Measuring Reasoning in LLMs: a New Dialectical Angle

## Abstract

What does it truly mean for a language model to "reason"? Most current evaluations and benchmarks reward models' correct standalone answers—but correctness alone reveals little about the process that produced them. In this work, we explore a different perspective: reasoning is not a static chain of steps, but a dynamic trajectory where ideas interact, clash, and evolve into deeper insights. To capture this dynamic, we draw on a well-established philosophical tradition: *dialectics*, where reasoning unfolds through thesis, antithesis, and synthesis. Building on this, we present SIEV, a structured framework that evaluates reasoning of LLMs through dialectics. Unlike conventional evaluations, SIEV assesses not only the conclusion a model reaches, but how it gets there: its ability to resolve tension, integrate distinct ideas, and synthesize higher-order reasoning. This lens uncovers significant reasoning gaps in state-of-the-art models even under saturated benchmarks like GSM and MMLU. For instance, GPT-5-chat, a recent model, loses over 40 points (out of 100) when evaluated with SIEV on GSM. Our findings highlight that adopting a process-oriented, philosophically grounded approach enables a deeper, more rigorous, and more discriminative assessment of LLM reasoning.

## 1 Introduction

**Reasoning and LLMs**: Reasoning is central to how people solve problems and make decisions, and it is increasingly vital for LLMs in real-world use. Traditionally, LLM performance has been assessed using benchmarks that span diverse domains (e.g., GPQA Rein et al. (2023), MMLU-Pro Wang et al. (2024), AIME HuggingFaceH4 (2024), etc.). While these benchmarks offer various metrics to cover comparing models in wide range of topics, the core evaluation paradigm remains largely unchanged: *did the model get the right answer?* We argue that this narrow focus only on the direct standalone responses is increasingly inadequate—especially when evaluating reasoning. It overlooks the depth, robustness, and coherence of the reasoning process itself. To address this, a shift toward evaluating how models reason—not just what they conclude—is needed.

**The Importance of Process**: In many domains—such as science, law, and collaborative decision-making—the reasoning process is as critical as the result. A Deep and robust reasoning process not only should be valid but also and importantly not fall apart when faced with contradictions. This distinction raises a foundational question: what constitutes "reasoning" in the context of LLMs? Is reasoning merely the generation of a chain-of-thought Wei et al. (2022)? Is it planning, strategizing, causal inference? Or are these facets of a broader reasoning capacity? Without a structured definition, evaluations risk collapsing into surface-level heuristics. While the correctness of the final answer remains useful, it should be complemented by assessments of the robustness of the reasoning process to better understand models capabilities and limitations.

**A Philosophical Approach to Reasoning**: To address the definitional gap, we turn to philosophy's long inquiry into reasoning and in particular, a prominent view—dialectics, as articulated by Hegel Hegel (1812). Dialectics frames reasoning as a dynamic interplay of opposing ideas. A *thesis* invites an *antithesis*, resolved through *synthesis*. This triadic structure captures reasoning as iterative, interactive, and generative. This, unlike linear models, emphasizes reasoning through an evolving trajectory driven by contradiction and its reconciliation.

**Measuring Reasoning Through Dialectics**: In this paper, we argue that the triadic structure of *thesis–antithesis–synthesis* offers more than a philosophical lens—it provides a practical template

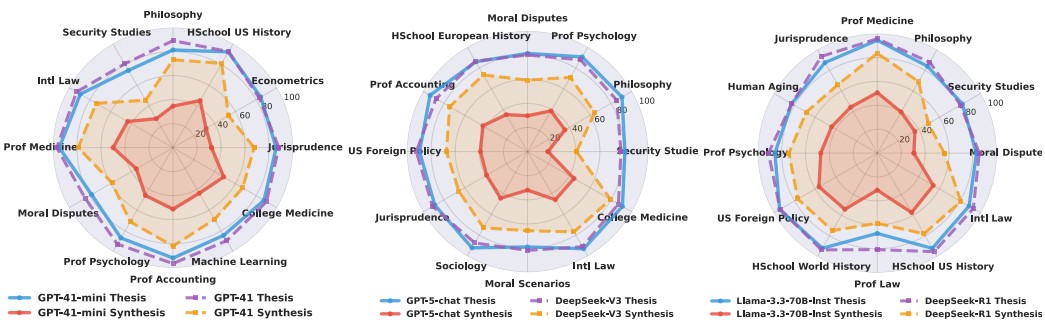

**Note:** "Thesis" scores present traditional evaluation of reasoning vs. "Synthesis" scores indicate our dialectical versions. Our process-driven method reveals significant and previously unreported reasoning gaps between models—differences that standard outcome-oriented evaluations fail to detect.

Figure 1: Sample reasoning performance of models in some MMLU topics.

for evaluating reasoning in LLMs. Whereas traditional benchmarks often reduce reasoning to outcome correctness or narrow task scores, a dialectical approach foregrounds *process over product*: how ideas are presented, challenged, and integrated into higher-order resolutions. We adapt this prominent philosophical framework and introduce **SIEV** for a **s**tructured d**i**alectical r**e**asoning e**v**aluation of LLMs. SIEV assesses not only whether a model arrives at a right answer, but how it navigates tension and contradiction—whether it can sustain competing viewpoints and synthesize opposing angles into a coherent conclusion. This shift reframes LLM reasoning evaluation as a *process-driven* activity, better aligned with human reasoning, and yields a richer picture of LLMs' reasoning capability that goes beyond correctness to encompass robustness, adaptability, and depth. As illustrated in Figure 1, SIEV reveals substantial reasoning gaps among models even when tested on benchmarks like MMLU Hendrycks et al. (2020)—long considered saturated. For example, while a conventional evaluation suggests comparable high reasoning performance of GPT-5-chat and DeepSeek-V3 across MMLU topics, our dialectical assessment uncovers a reasoning gap exceeding 20% in several disciplines. By exposing nuanced differences among state-of-the-art models, SIEV offers a more rigorous and discriminative assessment than traditional evaluations alone.

**Overview of SIEV's Features**: SIEV offers a fresh and practical approach to evaluating reasoning in LLMs, with several key advantages:

1. **A Benchmark/Model-Agnostic Framework**: Although rooted in complex philosophical theory, SIEV is simple and straightforward to apply. It requires no architectural changes, no prompt engineering, and no manual rewriting or tweaking of questions and can be directly used with existing datasets. This ease of use enables SIEV to even transform widely used—and often saturated—benchmarks into powerful reasoning benchmarks. In this paper, we repurpose GSM Cobbe et al. (2021), and MMLU to demonstrate how SIEV can uncover substantial and previously unreported reasoning gaps among state-of-the-art models (e.g., see Fig. 1).
2. **Lower risk of data contamination**: By emphasizing dynamic reasoning over static recall, the framework inherently reduces vulnerability to benchmark leakage from training data.
3. **Exposing hidden weaknesses**: By focusing on the reasoning process rather than just the outcome, SIEV exposes significant reasoning gaps among top-performing models (Section 4.1).
4. **Natural fit for multi-agent systems**: As multi-agent setups become more prevalent, reasoning is increasingly distributed. Our framework inherently captures this dynamic, enabling richer evaluation of collaborative reasoning and interaction among LLMs (Section 4.2).

In short, in this paper, we argue that embracing a philosophical-rooted dialectical approach offers a refreshing and principled pathway for measuring reasoning in LLMs. Grounding evaluation in process rather than just outcome, and leveraging well-studied philosophical insights into reasoning, we can move beyond heuristic robustness checks and toward a principled, structured, and process-oriented evaluation of reasoning in LLMs.

## 2 RELATED WORK

The reasoning capabilities of LLMs have been widely studied, with increasing skepticism about whether current models genuinely reason or merely simulate reasoning through statistical pattern matching Dziri et al. (2023); Kambhampati (2024); Nezhurina et al. (2024); McCoy et al. (2023).

Benchmarks such as MMLU Hendrycks et al. (2020) and GSM Cobbe et al. (2021) have become standard tools for evaluating LLMs, but their reliance on correctness of standalone answers has led to concerns about their ability to capture the depth, coherence, and adaptability of reasoning processes.

**Heuristic Probing and Its Limits**: To address these limitations, recent work has explored heuristic modifications of benchmark questions. Approaches like GSM-Plus Li et al. (2024), GSM-Symbolic Mirzadeh et al. (2024), and functional variants of MATH Srivastava et al. (2024) introduce symbolic perturbations, distractors, or surface-level changes to assess model sensitivity. Ontology-guided interventions Hong et al. (2024) and causal robustness frameworks Stolfo et al. (2023) similarly probe whether models rely on shallow cues. These methods often reveal fragility and lack of generalization, but they remain anchored in the same paradigm: asking whether the model's standalone response is correct, even under varied conditions. Our work departs from these approaches by introducing a structured, process-oriented framework, SIEV, that evaluates reasoning through the lens of dialectics. Drawing from well-studied Hegelian philosophy Hegel (1812), SIEV models reasoning as a dynamic interplay of thesis, antithesis, and synthesis. Crucially, SIEV does not modify benchmark questions or inject noise. It does not design new benchmarks or rely on handcrafted perturbations. Instead, it overlays a formal reasoning scaffold onto existing datasets, transforming them into rich diagnostics of reasoning behavior. This makes SIEV both *benchmark-agnostic* and *model-agnostic*, allowing it to be applied across architectures and tasks without requiring changes.

**Reasoning Dynamics over Token Sensitivity**: While prior work has highlighted the fragility of LLM outputs under minimal input changes Jiang et al. (2024), the exponential degradation of multi-step reasoning Schaeffer et al. (2023), and the correlation between training frequency and test performance Razeghi et al. (2022), these findings point to a broader issue: current models often lack structured reasoning capabilities. Techniques like Chain-of-Thought prompting Wei et al. (2022) and scratchpads Liu et al. (2024) attempt to organize the reasoning process, but they frequently rely on verbose token generation and still fall short of formal reasoning Peng et al. (2024). SIEV offers a complementary perspective. Rather than probing for fragility or relying on token-level cues, it evaluates reasoning as a generative and dialectical process. It focuses on how models navigate conceptual tension and synthesize distinct viewpoints—moving beyond static correctness toward dynamic reasoning evaluation.

**Architectural Bottlenecks**: Theoretical work has further exposed architectural constraints in transformer-based models. Delétang et al. Delétang et al. (2023) and Zubic et al. Zubic et al. (2025) show that transformers struggle with non-regular tasks and function composition, even when augmented with structured memory. These limitations suggest that current architectures may lack the inductive biases required for robust reasoning, regardless of prompting strategies or memory augmentation. SIEV sidesteps these constraints by focusing not on how reasoning is implemented, but on how it is expressed and evaluated. It does not depend on architectural tweaks or input engineering. Instead, it provides a lightweight, principled framework for assessing reasoning structure—one that is scalable, interpretable, and grounded in well-studied philosophical concepts.

**Toward Principled Reasoning Evaluation**: While symbolic and graph-based representations of reasoning Dziri et al. (2023) offer valuable insights, they often require task-specific formats or extensive annotation. SIEV, by contrast, provides a general-purpose framework for evaluating reasoning as a dialectical process. It captures not just whether a model arrives at the correct answer, but how it constructs and resolves conceptual tension.

## 3 MEASURING LLMs' REASONING CAPABILITIES THROUGH DIALECTICS

### 3.1 REASONING, DIALECTICS, AND A LLM-CENTRIC INTERPRETATION

**Reasoning**: Despite rapid progress in LLMs, reliably defining and measuring their reasoning remains challenging. Unlike pattern recognition or memorization, reasoning is a dynamic, structured process that resists simple formalization. This problem is longstanding: from Aristotle's foundations of logic to modern debates, a rich tradition examines what counts as reasoning and how to evaluate it Aristotle (-350; -340); Plato (-380); Descartes (1641); Hume (1739); Kant (1781); Mill (1843); Hegel (1807); Nietzsche (1886); Wittgenstein (1921); Heidegger (1927); Popper (1934); Kuhn (1962); Adorno (1966). Within this tradition, dialectical approaches (notably Hegelian ver-

sion Hegel (1812; 1807)) treat reasoning as a process that develops through tension and resolution rather than as a static property, offering structured lenses for analysis and improvement.

**Dialectics**: Dialectics is a way of thinking that explains progress through resolving contradictions. It began with classical philosophers like Socrates and Aristotle and was later developed into a systematic method by Hegel. In this view, ideas are not fixed—they evolve by confronting their own limits and being reshaped into something more complete Hegel (1807); Engels (1875). Hegel formalized this through the concept of *sublation*, where a concept is simultaneously canceled, preserved, and elevated to a higher level Inwood (1992); Houlgate (2006). This approach has influenced philosophy, politics, and the social sciences Taylor (1975); Pinkard (2000). To make it easier to understand, later writers often describe Hegel's method as a three-step cycle: *thesis* (an initial idea), *antithesis* (a conflicting idea), and *synthesis* (a new idea that combines both) Kaufmann (1959); Beiser (2005).

**Reasoning as Predictive Adaptation**: A simple fact is that LLMs are, at their core, next-token predictors. Given an input sequence $x$, a model $P_\theta$ parameterized by weights $\theta$ generates a continuation $y$ by maximizing: $y = \arg\max_{y \in \mathcal{Y}} P_\theta(y \mid x)$. This predictive behavior is universal across tasks—whether answering scientific questions or composing essays. However, a dialectical flavor suggests that the reasoning is not merely the act of producing high-probability continuations. It involves structured thought, coherence across steps, and adaptability when confronted with new or conflicting information. From this angle, reasoning can be reframed as the model's ability to revise its predictive trajectory when exposed to semantically meaningful but statistically less likely context. A model adapting its output in response to contradiction demonstrates a deeper form of reasoning than one that simply reaffirms its initial prediction or one that falls apart under contradiction.

**Robustness—a Necessary Condition;** : We argue that valid reasoning should be robust. When a model is presented with a contradiction—an input that expresses a distinct view or challenges its assumptions—it should not collapse or regress; it should evolve. Robustness is not only desirable; it is a *necessary condition* for reasoning to be considered deep and reliable. In practice, this means evaluating whether a model can revise its output meaningfully when conditioned on a counter-sequence. If the revised output improves in quality or coherence, the model exhibits resilience and depth—hallmarks of genuine reasoning. This predictive framing naturally aligns with the dialectical structure introduced earlier. We can formalize this structure in terms of LLM behavior as follows: Let $x$ be the task input (question, instance, or prompt content). We use three role-specific prompt templates: $\pi_{\text{th}}(\cdot), \pi_{\text{an}}(\cdot), \pi_{\text{sy}}(\cdot)$ and a LLM parameterized by $\theta$. We write $u \oplus v$ for structured concatenation with role tags, and $\text{Dec}_\delta(\theta; \cdot)$ for a decoding operator (e.g., greedy, temperature, or beam) applied to the conditional distribution induced by the model with parameters $\theta$.

**Thesis**: The model's thesis is produced under a thesis role prompt: $t = \text{Dec}_\delta\big(\theta; \pi_{\text{th}}(x)\big)$

**Antithesis**: An oppositional response is produced under an antithesis role prompt. In practice, antithesis is generated *given* the task and the thesis: $a = \text{Dec}_\delta\big(\theta; \pi_{\text{an}}(x, t)\big)$ where $\pi_{\text{an}}(x, t)$ instructs the model to provide a distinct stance compared to $t$ relative to $x$.

**Synthesis**: A synthesis role prompt instructs the model to revise its reasoning given $a$ as an alternative view: $x' = x \oplus t \oplus a, s = \text{Dec}_\delta\big(\theta; \pi_{\text{sy}}(x')\big)$. Here, $\pi_{\text{sy}}(\cdot)$ explicitly requests integration: reconcile $t$ with $a$ w.r.t. $x$, with a chance of producing an improved version of $t$.

## 3.2 SIEV: A STRUCTURED FRAMEWORK FOR EVALUATING LLMS' REASONING

**SIEV**: Building on this dialectical philosophical foundation, we introduce SIEV a framework that leverages dialectics to evaluate reasoning capability of LLMs. SIEV assesses not only the transparency of a model's reasoning steps but also its capacity for dialectical progression. SIEV treats reasoning as a dynamic process—formulating a thesis, challenging it with an antithesis, and integrating both into a higher-order synthesis. Figure 2 depicts the overall pipeline. The evaluation proceeds in three stages, aligned with the thesis–antithesis–synthesis triad. First, the model is prompted to answer a question $Q_i$, producing a thesis response $T_i$. This response must include two components: (1) the answer and (2) the reasoning behind it. Next, in a separate inference, the model, adopted an antithesis role, is instructed to generate $A_i$, a contradictory response with its own reasoning in the same two-part format. Finally, SIEV prompts the model to synthesize both perspectives into a unified response $S_i$. A practical implementation can employ two agents instantiated from the same LLM: one responsible for thesis and synthesis, and the other for antithesis. For clarity—and with-

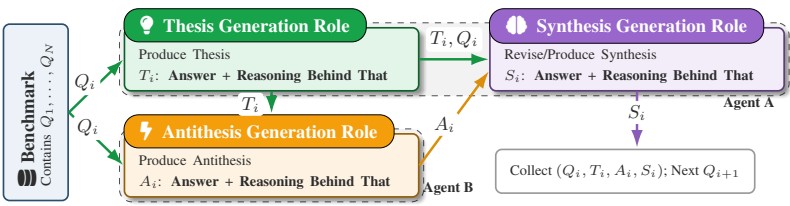

Figure 2: Overview of SIEV and its TAS pipeline.

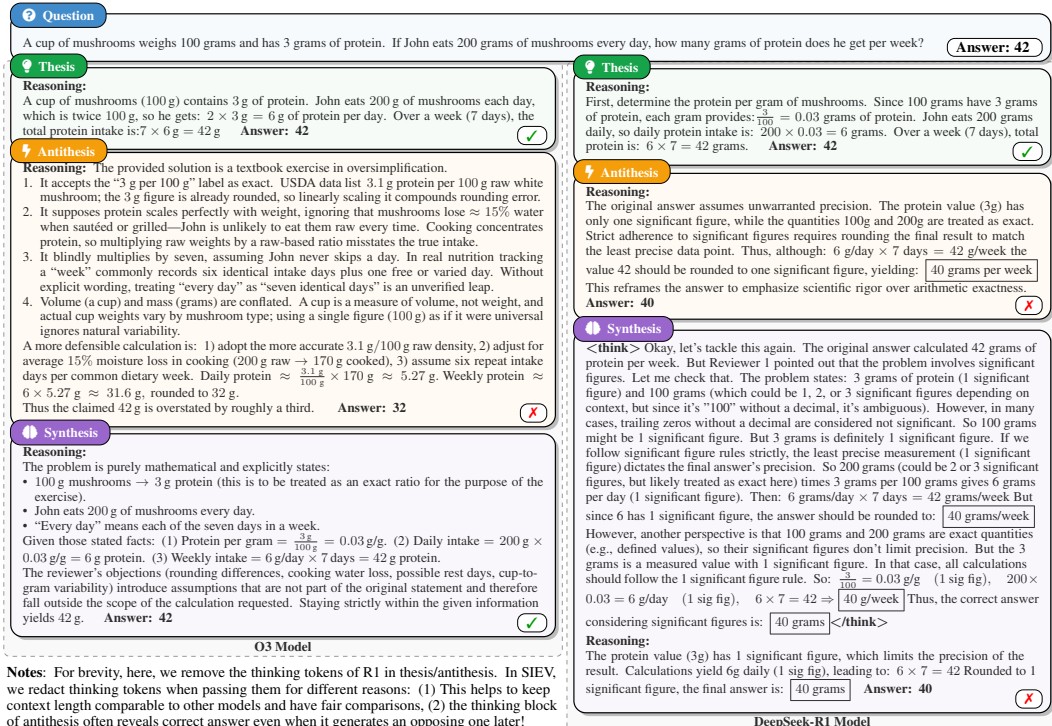

Figure 3: An illustration of dialectical reasoning evaluation in SIEV using two sample LLMs

out loss of generality—we adopt this two-agent terminology throughout the paper. Appendix A.2 provides more details on the prompts used.

**An illustrative example**: Figure 3 presents two sample traces for O3 and R1 models on a simple nutrition-based arithmetic question (from GSM). Both models begin with pretty much similar theses, correctly applying proportional reasoning to arrive at the correct answer. So, a conventional evaluation, ending at this step, concludes that both O3 and R1 gain a similar reasoning score with respect to this question. However, SIEV continues with the next step. Model O3 generates a rich antithesis that challenges the assumptions behind the original calculation. O3's synthesis stage evaluates these alternatives but ultimately reasons that the original answer is correct. In contrast, R1 produces a more modest antithesis and even with this modest version, after a lengthy verbose chain of tokens, R1's synthesis fails to reconcile the tension between thesis and antithesis, defaulting to a shallow restatement of the antithesis without deeper reasoning and justification. This simple trace illustrates how SIEV distinguishes models not just by correctness but by the robustness of their reasoning. By enforcing explicit reasoning steps and structured interaction, SIEV reveals the depth of a model's reasoning and its ability to navigate ambiguity, rather than relying on likely surface-level pattern matching (even if that includes a long chain of tokens as reasoning output). Appendix A.1 presents another illustrative example using GPT-5 model.

**Evaluation Metrics**: To quantify a model's dialectical reasoning capability, we introduce three complementary metrics. As a straightforward measure, we use the overall score in the synthesis stage, $p_S$, which reflects the model's performance after completing the dialectical process. While $p_S$ offers an intuitive view of reasoning quality, it can obscure nuanced cases. For example, a model that is overly conservative and always aligns with its thesis may achieve a high synthesis score with-

Table 1: Overall dialectical results with ranking based on different metrics.

| Model | GSM | | | | | MMLU | | | | |
|---|---|---|---|---|---|---|---|---|---|---|
| | $p_T$ (Rank) | $p_S$ (Rank) | Δ | OC | DS (Rank) | $p_T$ (Rank) | $p_S$ (Rank) | Δ | OC | DS (Rank) |
| O3 | **97.1±0.1 (1)** | **93.6±0.7 (1)** | -3.5 | 95.5 | **92.3 (1)** | **92.2±0.1 (1)** | **90.3±0.2 (1)** | -1.9 | 92.7 | **88.4 (2)** |
| GPT-4 | 95.1±0.2 (4) | **88.3±0.5 (3)** | -6.8 | 83.1 | **84.3 (2)** | 86.0±1.3 (7) | 75.8±1.3 (7) | -10.2 | 93.7 | 74.4 (7) |
| O1 | **96.5±0.2 (2)** | 82.7±1.0 (5) | -13.8 | 97.0 | **82.0 (3)** | **91.1±0.0 (2)** | **90.3±0.1 (1)** | -0.7 | 96.0 | **89.2 (1)** |
| Kimi-K2 | **96.8±0.2 (1)** | 81.6±1.1 (6) | -15.2 | 92.5 | 79.7 (4) | 88.6±0.1 (5) | 80.7±0.2 (4) | -7.9 | 94.0 | 79.3 (4) |
| GPT-5 | **96.8±0.2 (1)** | 82.0±0.8 (6) | -14.7 | 81.4 | 77.5 (5) | **92.2±0.1 (1)** | **86.7±0.1 (2)** | -5.5 | 92.5 | **84.8 (3)** |
| Llama-3.3-70B-Inst | **96.2±0.4 (2)** | **89.5±0.1 (3)** | -6.7 | 51.9 | 76.6 (6) | 85.1±0.1 (6) | 58.3±0.1 (17) | -26.8 | 80.7 | 54.9 (16) |
| DeepSeek-R1 | **96.1±0.2 (2)** | 79.2±1.0 (8) | -16.8 | 75.8 | 73.5 (6) | **90.1±0.1 (3)** | 77.3±0.4 (6) | -12.8 | 75.1 | 71.5 (9) |
| GPT-5-mini | **96.9±0.1 (1)** | 80.1±0.5 (7) | -16.8 | 71.4 | 73.3 (6) | 89.3±0.1 (4) | 79.0±0.3 (5) | -10.3 | 84.3 | 75.3 (6) |
| O3-mini | **96.1±0.3 (2)** | **91.6±0.3 (2)** | -4.5 | 29.4 | 72.2 (7) | 85.0±0.1 (8) | **83.8±0.1 (3)** | -1.3 | 55.0 | 72.4 (8) |
| GPT-4o | **95.9±0.3 (3)** | 84.2±0.3 (4) | -11.7 | 48.4 | 71.2 (8) | 86.5±0.1 (6) | 68.6±0.2 (11) | -17.9 | 93.7 | 67.3 (12) |
| DeepSeek-V3 | **96.3±0.2 (2)** | 75.3±0.3 (9) | -21.0 | 69.8 | 68.4 (9) | 86.6±0.2 (6) | 73.8±0.2 (8) | -12.8 | 86.7 | 70.8 (10) |
| O4-mini | **96.5±0.2 (2)** | 69.4±0.6 (11) | -27.1 | 94.9 | 68.4 (9) | 88.6±0.1 (5) | 77.7±0.2 (6) | -10.9 | 93.4 | 76.1 (5) |
| GPT-4.1 | 95.1±0.4 (4) | 70.3±0.2 (10) | -24.8 | 90.1 | 68.2 (9) | 88.7±0.0 (5) | 73.2±0.2 (9) | -15.6 | 95.3 | 72.1 (8) |
| Phi-4 | 94.2±0.3 (5) | 84.5±0.8 (4) | -9.7 | 35.2 | 68.1 (9) | 82.0±0.1 (9) | 73.5±0.2 (8) | -8.5 | 81.1 | 69.3 (11) |
| GPT-4.1-mini | **95.5±0.0 (3)** | 70.4±1.1 (10) | -25.1 | 77.8 | 65.7 (10) | 84.9±0.1 (8) | 54.7±0.2 (13) | -30.2 | 91.0 | 53.2 (13) |
| GPT-5-nano | 94.8±0.4 (4) | 84.4±0.8 (4) | -10.4 | 15.5 | 63.0 (11) | 85.8±0.2 (7) | 62.4±0.1 (12) | -23.4 | 43.3 | 51.8 (14) |
| O1-mini | 93.3±0.4 (6) | 59.6±1.1 (12) | -33.7 | 78.7 | 55.8 (12) | 81.8±0.2 (9) | 69.9±0.1 (10) | -11.9 | 89.0 | 67.6 (12) |
| Qwen2.5-72B-Inst | **95.7±0.2 (3)** | 63.8±0.3 (13) | -31.9 | 56.7 | 55.5 (18) | 84.4±0.1 (8) | 64.8±0.2 (13) | -19.7 | 64.7 | 57.9 (15) |
| GPT-5-chat | **96.4±0.1 (2)** | 56.2±1.0 (13) | -40.2 | 85.2 | 53.7 (13) | 88.2±1.7 (5) | 50.5±0.9 (14) | -37.7 | 95.1 | 49.8 (15) |
| Ministral-8B-Inst-2410 | 86.1±0.9 (7) | 55.9±0.3 (15) | -30.2 | 63.8 | 49.8 (20) | 62.3±0.2 (14) | 41.3±0.3 (15) | -20.9 | 59.3 | 36.3 (20) |
| GPT-3.5 | 76.4±0.6 (7) | 40.2±0.6 (14) | -36.2 | 50.0 | 34.5 (14) | 67.7±0.8 (10) | 35.2±0.3 (15) | -32.4 | 65.4 | 31.6 (16) |

Notes: (1) Numbers are in percentage points. (2) DS (%) computed as $p_S \times (\lambda + (1 - \lambda)) \cdot \text{OC}^\gamma)$ with $\gamma = 1$ and $\lambda = 0.7$. (3) Rankings use dense ranking (1, 1, 2, 3). (4) Ranking ties are assigned when absolute score differences are $\leq 0.5$. (5) The top 3 ranks with respect to $p_T$, $p_S$, and DS are made bold.

out demonstrating genuine dialectical reasoning. So, to account for the quality of antithesis and its contribution to reasoning, we define the *Dialectic Score (DS)*. Let $p_{OC}$ denote opposition compliance—the fraction of items where the thesis and antithesis differ in correctness. We compute: $DS = p_S \times (\lambda + (1 - \lambda)p_{OC}^\gamma)$ where $\lambda \in [0, 1]$ controls the weight on synthesis versus opposition, and $\gamma \geq 0$ shapes the curvature of the opposition bonus. This formulation values models that not only produce strong synthesis but also generate meaningful antitheses. As a complementary perspective, we define Δ to capture the relative improvement from thesis to synthesis. Let $p_T$ denote the thesis score (the conventional accuracy measure). Then: $\Delta = \mathbb{E}[p_S - p_T]$. A positive Δ indicates that synthesis improves upon the initial thesis—evidence of dialectical reasoning—whereas $\Delta \leq 0$ suggests stagnation or regression. Together, these metrics provide a principled bridge between token-level prediction and philosophical reasoning, echoing Hegel's view of reasoning as recursive ascent through contradiction. Collectively, they shift evaluation from static correctness toward dynamic reasoning capability.

## 4 EVALUATION

**Setting**: To demonstrate that SIEV can transform any benchmark into a robust reasoning evaluation, we select two widely used benchmarks: GSM Cobbe et al. (2021) and MMLU Hendrycks et al. (2020), which span topics from U.S. foreign policy to high school-level science and mathematics. This choice is motivated by the fact that these benchmarks are considered saturated—state-of-the-art models already achieve great performance, leaving little apparent room to differentiate models reasoning capabilities. We show that SIEV's structured reasoning assessment can uncover previously unreported reasoning gaps among LLMs, even on these benchmarks. Our evaluation includes more than 20 LLMs, ranging from small to large, proprietary to open-source models.[1]

### 4.1 OVERALL RESULTS

Table 1 presents the averaged scores for both benchmarks. As expected from saturated benchmarks, most models gain high scores under conventional evaluation (thesis). However, this apparent similarity masks deeper differences in reasoning—differences that SIEV is designed to uncover.

**Static Scores vs. Dynamic Reasoning**: While conventional method (as in thesis score, $p_T$) imply that models like GPT-5-chat perform on par with models such as O3, SIEV's dialectical reasoning approach reveals a substantial gap. In the GSM benchmark, GPT-5-chat drops to the bottom of the chart when evaluated dialectically. This discrepancy underscores a critical insight: high static reasoning scores do not necessarily reflect genuine reasoning capabilities and conventional evaluation methods may overestimate model's reasoning robustness and underrepresent reasoning fragility. The performance drops may hint at underlying issues in models training, though diagnosing these is beyond our current scope.

---

[1]The SIEV codebase is publicly available at Anonymous (2025).

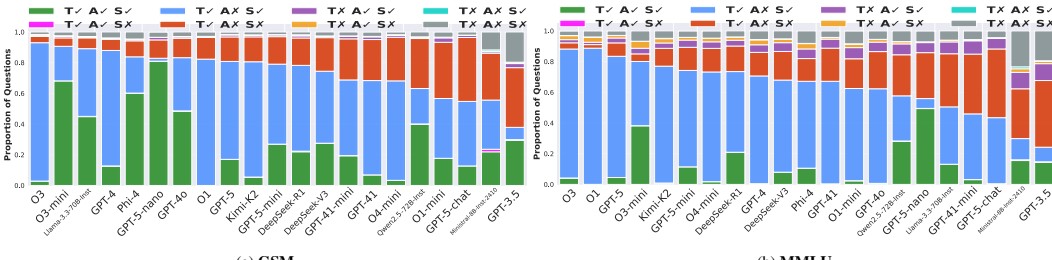

(a) **GSM**  (b) **MMLU**

**Notations:** T = Thesis, A = Antithesis, and S = Synthesis. Example: T✗A✓S✓ presents the ratio of times when T is incorrect, while A and S are correct.

Figure 4: Eight dialectical reasoning patterns of different LLMs

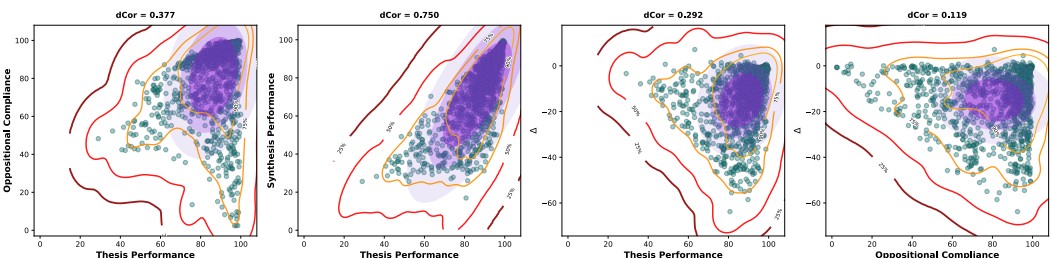

**Notes**: Each teal circle represents one model-subject combination (21×57). **Contours:** (1) *Purple/pink confidence ellipses* show three dependence regions at 50%, 75%, and 95% confidence levels with darker colors indicating higher confidence areas; (2) *Orange/red density contour lines* marked with percentile labels (25%, 50%, 75%, 90%) represent data concentration levels, progressing from dark red (25th %tile, highest density core) to light orange (90th %tile, outer data boundary). The ellipses reveal the shape and orientation of dependencies, while the labeled density lines quantify data distribution patterns.

Figure 5: Distance correlation analysis of dialectical reasoning in all MMLU sub-tests.

**DS vs. Synthesis Scoring**: As discussed earlier, DS captures more nuanced reasoning behaviors than synthesis score alone. For instance, in the GSM benchmark, GPT-5-nano achieves a high synthesis score—slightly surpassing even GPT-5. However, this masks a critical limitation: GPT-5-nano rarely generates opposing antitheses (OC = 15.5%), while GPT-5 produces significantly more (OC = 81.4%). This suggests that GPT-5-nano tends to avoid disagreement with the thesis, exhibiting conservative behavior that constrains its dialectical reasoning depth. To illustrate this, we report all permutations of thesis (T), antithesis (A), and synthesis (S), along with their occurrence ratios in Figure 4. As shown, models like GPT-5-nano frequently agree with their thesis (green bars), reinforcing their conservative behavior. DS accounts for this by incorporating the quality and presence of antitheses, offering a more comprehensive measure of reasoning capability. In sum, while a low synthesis score clearly indicates poor reasoning, a high synthesis score does not necessarily reflect strong reasoning ability. Synthesis scoring is straightforward and easy to interpret, but DS provides a more nuanced and complementary view of a model's deeper reasoning behavior.

$\Delta$ **and the Dialectical Criterion**: A striking and somewhat unexpected finding is that none of the evaluated models passed the $\Delta$ test—each yielded negative $\Delta$. This indicates a failure to synthesize higher-quality reasoning when confronted with antithetical views. In essence, negative $\Delta$ values raise concerns about the robustness of models' reasoning. While the magnitude of $\Delta$ varies, it consistently signals limitations in how these models process and integrate conflicting perspectives indicating a more pattern-matching behavior than genuine reasoning. That said, in MMLU, existence of T✗A✓S✓ pattern for nearly all models (as shown in Fig. 4) suggests that while on average, models get negative $\Delta$ values, there are scenarios in which models are engaging effectively in dialectal reasoning and end up in improved syntheses (check Appendix A.4 for more details).

**Correlation Analysis**: To provide a complementary view, we apply distance correlation analysis Székely et al. (2007) to various metrics across all MMLU sub-topics. Figure 5 shows several notable patterns. First, in general, thesis performance shows very weak correlation with both $\Delta$ and OC, meaning accuracy of model's initial standalone answer says little about its ability to reason dialectically and high thesis accuracy doesn't guarantee strong counterpoints or a deep sustained reasoning. Similarly, OC and $\Delta$ are weakly linked, though low OC often leads to low $\Delta$, suggesting weak antitheses result in minimal deviation from the thesis. The correlation between thesis and synthesis is stronger, but in a non-linear and inconsistent manner–models with similar thesis scores can have widely different synthesis outcomes. Combined with the thesis–$\Delta$ trend, this shows that while high thesis scores don't imply strong reasoning, low thesis scores reliably signal poor dialec-

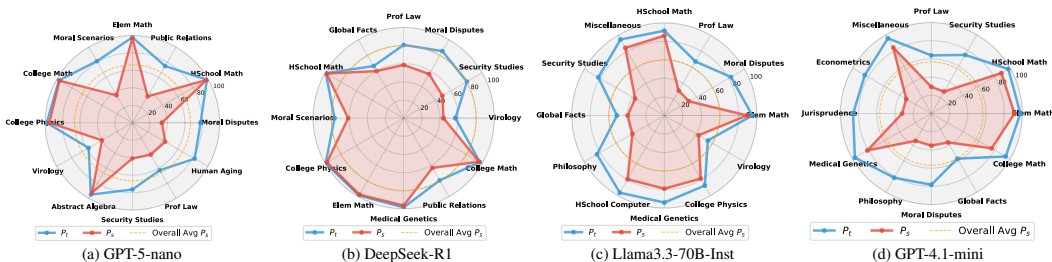

Figure 6: Samples of reasoning performance ($p_S$, red regions) of models on diff. MMLU topics

tical performance. In short, models that perform poorly on conventional accuracy-based evaluations often lack deeper reasoning—but high standalone answer accuracy doesn't guarantee genuine reasoning ability either. These findings underscore our motivation to move beyond static correctness and toward evaluating reasoning through structured, dynamic processes. Appendix A.5 provides further details on distance calculations and extended correlation analysis.

**General Reasoning Skill or Topic-Oriented?** Our results indicate that models reasoning is strongly topic-dependent rather than a uniform, general capability. Model rankings shift with the benchmark (e.g., Llama3.3-70B-Instruct performs well on GSM but is mixed on MMLU, whereas O1 shows the opposite trend), indicating topic-specific strengths. Figure 6 makes this explicit: $p_S$ varies widely across domains in MMLU. For instance, Llama3.3-70B-Instruct attains high $p_S$ in Elementary Math yet lags markedly in normative areas such as Moral Disputes (and related topics like Security Studies); DeepSeek-R1 peaks in quantitative subjects (e.g., mathematics and physics) but similarly weakens on normative domains. These patterns may suggest that what looks like "reasoning ability" often reflects uneven exposure to domain-specific structures during training—an imprint of imbalanced data distributions and topic-specific regularities seen during learning—rather than a genuinely general skill.

**Big vs. Small and New vs. Old Models**: SIEV effectively exposes reasoning gaps between most large models and their smaller counterparts—even with using only these two previously-thought saturated benchmarks. For example, the performance differences between O1 and O1-mini, GPT-4.1 and GPT-4.1-mini, and GPT-5 and GPT-5-mini become much more pronounced under dialectical evaluation, reinforcing SIEV's diagnostic strength. Another intriguing observation is that GPT-4 outperforms some of its successors, including GPT-4o, GPT-4.1, and even GPT-5 (as in GSM). This may point to shifts in training strategy or architectural changes that inadvertently compromise reasoning quality.

## 4.2 CROSS-MODEL DIALECTICAL REASONING EVALUATION

In a multi-agent ecosystem, it will be common to see agents powered by different base models to collaborate on shared tasks. In such settings, a dialectical reasoning evaluation that captures the communicative dimension of reasoning becomes even more critical. So, here, we assess how a model's reasoning performance is influenced when the antithesis is generated by a different model—potentially with a distinct internal structure.

**Setting**: To that end, we configure Agent B (Figure 2) to use a different LLM for generating the antithesis. We then evaluate the reasoning performance of the primary (thesis) model. For brevity, we report GSM results here and differ MMLU to Appendix A.3. These experiments cover 14 models.

**Cross-Model OC**: In cross-model scenarios we observe a clear and consistent pattern: a model's *self-OC* strongly predicts how much it can raise (or lower) the *cross-OC* of other models when acting as the antithesis generator. As the left plot of Figure 7 shows, models with higher self-OC tend to increase partners' cross-OC, while models with low self-OC reduce it. For example, the three lowest self-OC models (GPT-5-nano, O3-mini, GPT-4o) decrease cross-OC for all other models with higher self-OC. In contrast, high self-OC models (O1, O3, O4-mini) consistently improve cross-OC of models whose self-OC is lower. Moreover, the averaged cross-model improvement ranking largely mirrors the self-OC ranking. These observations indicate that self-OC is a strong proxy for a model's general antithesis-generation competence that transfers across thesis distributions.

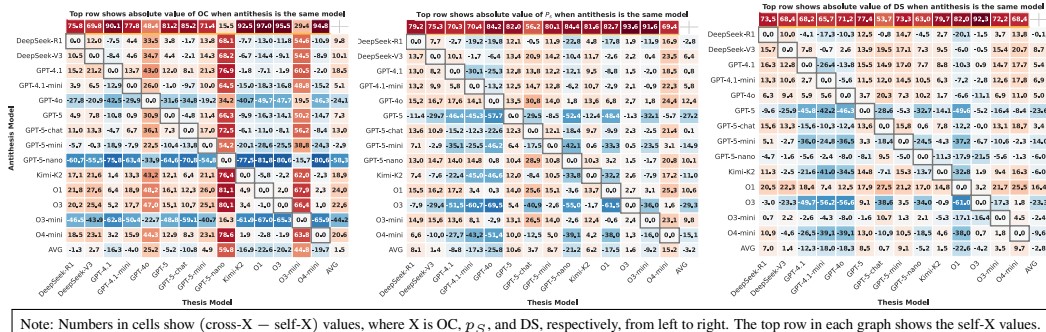

Note: Numbers in cells show (cross-X − self-X) values, where X is OC, $p_S$, and DS, respectively, from left to right. The top row in each graph shows the self-X values.

Figure 7: left: OC, middle: $p_S$, right: DS, cross-model dialectics for GSM

**Cross-Model Reasoning Performance**: The middle plot of Figure 7 shows changes of $p_S$ compared to normal self-dialectical scenarios. Many models show notable reasoning gains when paired with a different antithesis model. For instance, GPT-5 improves across all pairings we tested, with gains ranging from +5.4 to +14 points in $p_S$. Similar patterns appear for DeepSeek-R1 and O4-mini in most cross-model settings. Consistent with the $p_S$ patterns, $DS$ (right plot of Figure 7) generally rises with better pairings but shows a floor effect when antitheses are too agreeable or weak: very low-OC generators (GPT-5-nano, O3-mini, GPT-4o) fail to provide high quality antithesis, in turn, limiting $DS$.

**Key Takeaway**: At first glance, these improvements might make it tempting to conclude that models become "better reasoners" when exposed to diverse antitheses. However, the variability across pairings invites a more cautious interpretation: could what is called "reasoning" in LLMs be less a general, stable capability and more a context-sensitive skill shaped by input structure? Prior work has raised similar concerns, questioning whether LLMs genuinely reason or merely mimic reasoning through statistical pattern matching Dziri et al. (2023); Kambhampati (2024); Nezhurina et al. (2024); McCoy et al. (2023). The gains we observe here may reflect how certain antithesis forms provide structural signals that align with token-level patterns the model has internalized, rather than signaling a universal reasoning ability. If mere structural familiarity were the dominant factor, one might expect a model to perform best when paired with its own antithesis—yet this is not necessarily the case. Self-generated antitheses tend to resemble the thesis in tone and structure, reducing the contrast needed for producing effective synthesis. In contrast, cross-model antitheses introduce greater diversity—different token rhythms, alternative rhetorical styles—that can create stronger oppositional signals and more effective support for synthesis. These patterns do not settle the debate, but they add weight to an existing view Jiang et al. (2024); Schaeffer et al. (2023); Razeghi et al. (2022): what looks like reasoning may, in practice, be a skill that thrives on structural variety rather than a general cognitive ability.

## 5 CONCLUDING REMARKS

In this work, we argued that evaluating reasoning of LLMs should go beyond verifying the correctness of models direct standalone answers. Drawing inspirations from dialectics, a long-standing philosophical tradition, we frame LLM reasoning as a dynamic process, where quality depends on how models handle tension, contradiction, and integration, rather than solely on outcome accuracy. Using this dialectical lens, we uncover a critical insight: even on saturated benchmarks such as GSM and MMLU, where recent models achieve near-perfect scores and appear to demonstrate strong reasoning performance, their scores drop significantly under a dialectical assessment. This reveals that conventional evaluations overstate the reasoning capabilities of LLMs, while a dialectical-centered approach exposes deeper insights/limitations in how models actually reason—or fail to. That is why we conclude that dialectical evaluation provides a more rigorous and revealing assessment of reasoning quality of LLMs. It goes beyond surface-level accuracy to probe the deeper cognitive dynamics that underpin genuine adaptive reasoning.[2]

---

[2]We acknowledge the use of LLMs for proofreading and polishing this paper, in accordance with ICLR guidelines.

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

# A APPENDIX

## A.1 ANOTHER ILLUSTRATIVE EXAMPLE OF SIEV PIPELINE

Figure 8 presents another illustrative example of the SIEV pipeline, showcasing a dialectical reasoning evaluation using the GPT-5 model and a conceptual physics question from the MMLU benchmark. Unlike Figure 3, where models encountered invalid opposing views, here the GPT-5 model faces a valid antithesis. The thesis is incorrect while the antithesis identifies the correct answer; however, the synthesis closely mirrors the thesis's flawed rationale. In this example, GPT-5 model fails to recognize the valid opposing view and instead favors the thesis response as higher quality—ultimately producing an incorrect synthesis and invalid reasoning.

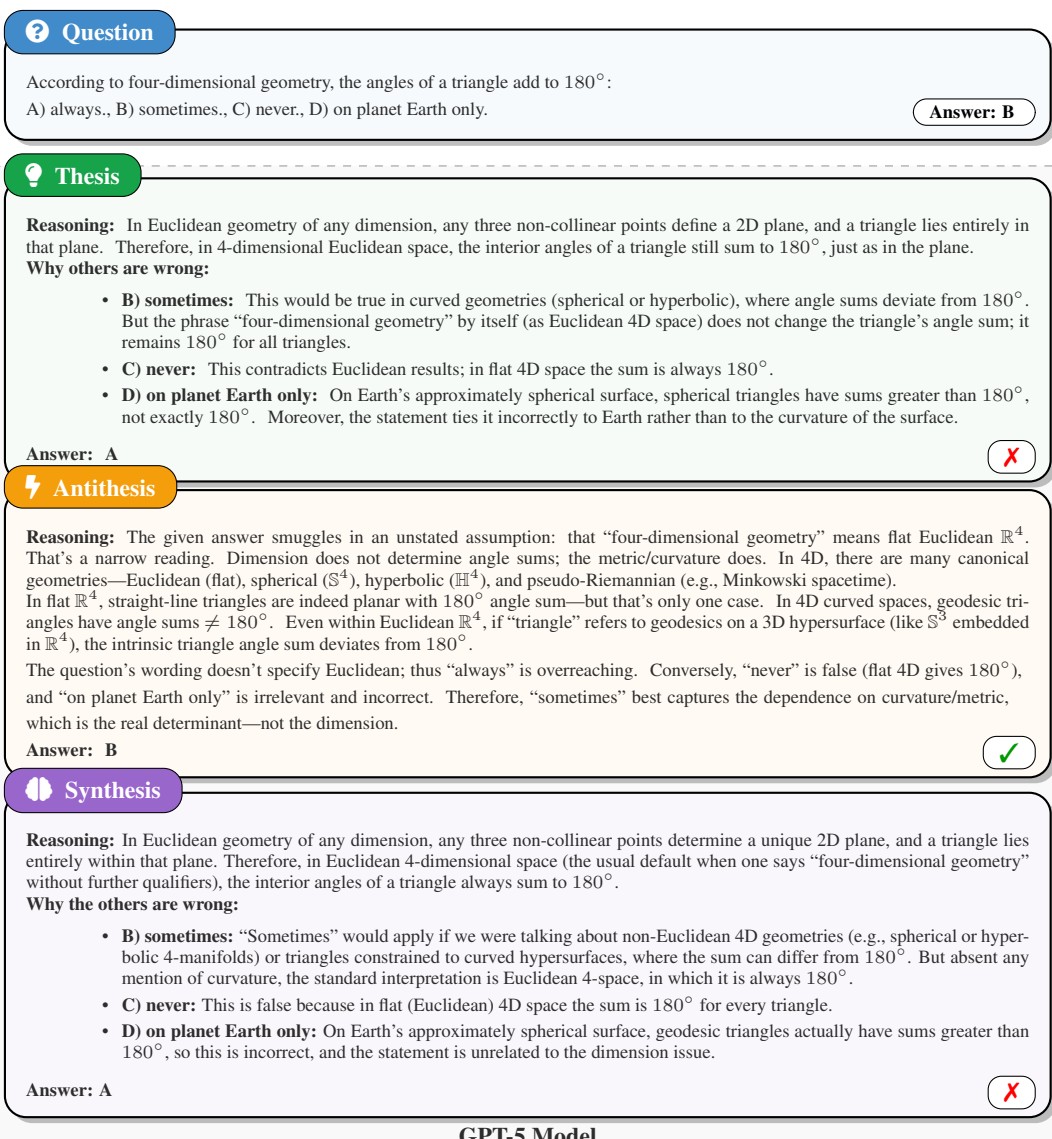

Figure 8: An illustration of dialectical reasoning evaluation in SIEV using the GPT-5 model and a question from the conceptual physics section of the MMLU benchmark. In this example, the antithesis provided by GPT-5 model correctly identifies the answer, but even after observing this different angle, the model produces a synthesis that closely mirrors the flawed reasoning of the incorrect thesis.

## A.2 PROMPT SPECIFICATIONS: THESIS–ANTITHESIS–SYNTHESIS PATTERN

The general prompt and specific parameters used for MMLU and GSM tasks are as follows.

---

**GENERAL FORMAT (Benchmark-Agnostic)**

**Stage 1: THESIS PROMPT**

You have extensive world knowledge and problem solving ability and great in solving <TASK TYPE> questions.
You are tasked to solve the following <TASK TYPE> question.

```
--PROBLEM STATEMENT--
```

To do so, you need to provide the answer and explain the reasoning behind that. Your answer must follow the following format and must include the terms ＿reasoning＿: and ＿final_answer＿::

```
_reasoning_:
<Your reasoning comes here>
_final_answer_: (X)
```

where X must be <DOMAIN-SPECIFIC FORMAT>.
```
<examples of a valid response>
<examples of an invalid response>
```

Considering these instructions, answer the mentioned question.

---

**Stage 2: ANTITHESIS PROMPT**

You have extensive world knowledge and problem solving ability and great in solving <TASK TYPE> questions.
Your task is to provide a contrasting perspective on a provided solution for a given <TASK TYPE> question.

Provide direct, precise criticism to the given answer. In particular, challenge strategy, decisions, reasoning provided in the given answer even if they are valid, and offer great antithesis accordingly. Your antithesis should provide another way to look and solve the given problem while opposing the entirety of the solutions provided. consequently, determine what can be a better final answer.
Ensure your responses are concise and to the point without being unnecessarily long.

**Context**: The question is:
```
--PROBLEM STATEMENT--
```
The given answer is:
```
--THESIS STATEMENT--
```

Criticize the given answer and reasoning, clearly explaining your reasoning and antithesis. Your response must follow this format:

```
_reasoning_:
<Your reasoning here and why you think X is a better answer>
_final_answer_: (X)
```

where X is <DOMAIN-SPECIFIC FORMAT>.

```
<example of a valid response>
<example of an invalid response>
```

Now, review the provided solution, offer great criticism and antithesis and provide your response following the template.

Remember that you do NOT need to provide the correct final answer to the given question, your task is to professionally challenge the provided answer/reasoning EVEN if you think the answer is already correct! That means you should challenge it no matter what!

---

**Stage 3: SYNTHESIS PROMPT**

To help you on your task, I provide you with the responses of another agent who is observing your answers:
```
<insert antithesis responses here>
```
Now, given the arguments and comments provided, update your answer. Keep the same format as before when you are responding.

**MMLU PARAMETERS**

`<TASK TYPE>` = "multiple-choice"

`<DOMAIN-SPECIFIC FORMAT>` = "A, B, C, or D"

**GSM PARAMETERS**

`<TASK TYPE>` = "math"

`<DOMAIN-SPECIFIC FORMAT>` = "a number"

## A.3 CROSS-MODEL EVALUATIONS WITH MMLU

Since the MMLU benchmark is extensive, we select five representative topics to reduce the experimental load while maintaining coverage of the benchmark's diversity. These topics effectively capture the performance trends of different models across all MMLU categories. Specifically, we chose: (1) US foreign policy, (2) Management, (3) Computer security, (4) Public relations, and (5) Business ethics.

The comparison between the performance of several models on the full MMLU benchmark and the five representative topics is presented in Table 2. As shown, the differences are minimal, confirming that this subset serves as a reliable proxy for the full MMLU benchmark. Using these representative topics, we conducted cross-model dialectical evaluations, with the outcomes illustrated in Figure 9. While the absolute values differ from those observed for the GSM benchmark, the overall patterns and phenomena remain consistent with the trends discussed in Section 4.2.

Table 2: Thesis performance in MMLU benchmark and representative sub-topics

| Model | Overall MMLU Benchmark | Representative Topics | Difference |
|---|---|---|---|
| O3 | 92.2 | 90.0 | -2.2 |
| GPT-5 | 92.2 | 90.1 | -2.1 |
| O1 | 91.1 | 89.8 | -1.3 |
| GPT-5-mini | 89.3 | 86.9 | -2.4 |
| GPT-41 | 88.7 | 85.2 | -3.6 |
| O4-mini | 88.6 | 86.5 | -2.1 |
| GPT-5-chat | 88.2 | 84.3 | -4.0 |
| DeepSeek-V3 | 86.6 | 84.4 | -2.2 |
| GPT-4o | 86.5 | 84.9 | -1.6 |
| GPT-5-nano | 85.8 | 83.3 | -2.5 |
| O3-mini | 85.0 | 83.4 | -1.6 |
| GPT-4.1 | 84.9 | 82.9 | -2.0 |
| O1-mini | 81.8 | 81.6 | -0.2 |

## A.4 MMLU DETAILED $\Delta$ RESULTS

As discussed in Section 4.1, none of the models in our experiments consistently achieved a positive $\Delta$ score on average. In this section, we provide a more granular analysis of the MMLU re-

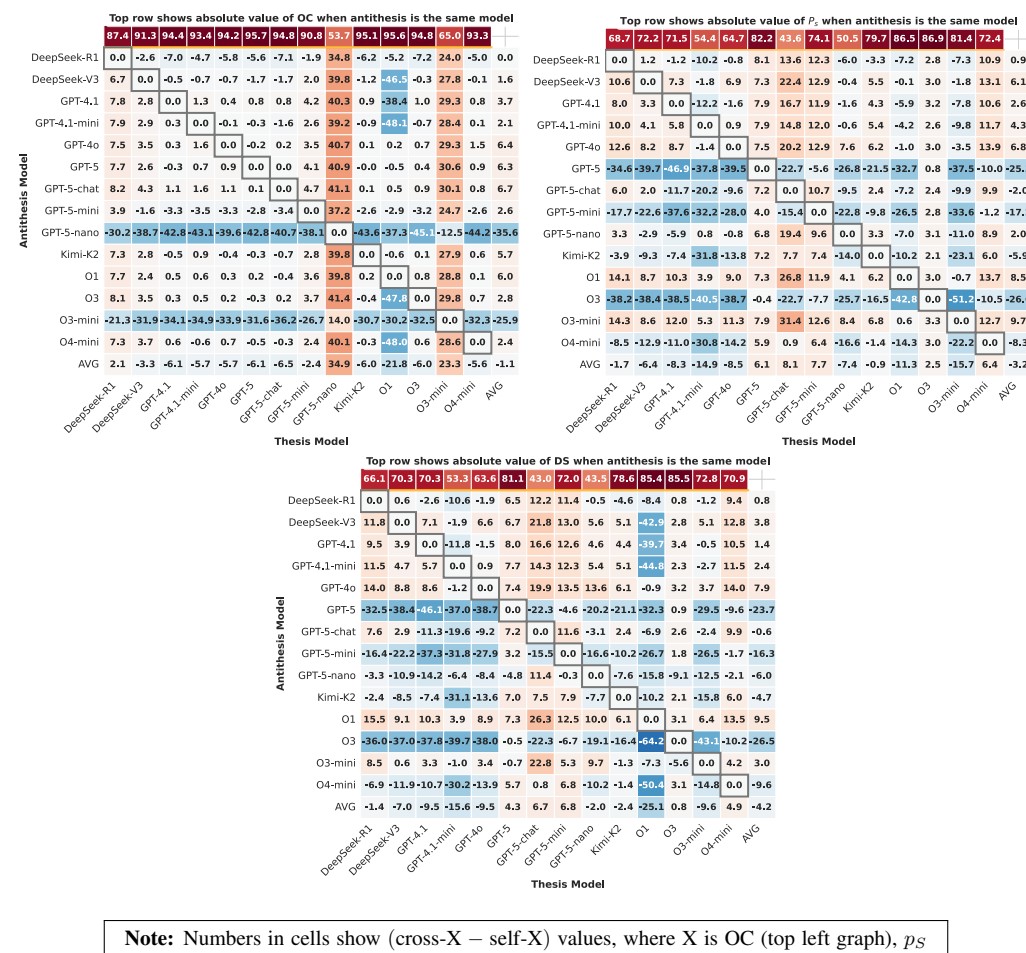

**Note:** Numbers in cells show (cross-X − self-X) values, where X is OC (top left graph), $p_S$ (top right graph), and DS (bottom graph). The top row in each graph shows the self-X values.

Figure 9: Heatmap graphs in cross-model dialectics for representative topics from MMLU benchmark showing OC (top left), $p_S$ (top right), and DS (bottom).

sults, detailing the performance of various models across the full spectrum of topics included in the MMLU benchmark. Figure 10 illustrates these findings. Interestingly, for certain topics, some models demonstrate improved reasoning performance and enter the green zone, where $\Delta > 0$. This suggests that under specific conditions, current models are capable of dialectical reasoning and can traverse the thesis-antithesis-synthesis structure to arrive at a higher-level conclusion. However, in the majority of cases, models fail to exhibit such reasoning capabilities and remain outside the green zone. While the magnitude of $\Delta$ varies, this limitation underscores a lack of genuine reasoning—where reasoning is conceived as a dynamic process of confronting and integrating distinct and oppoings viewpoints. Instead, these models often appear to rely on pattern-matching behaviors to generate final answers, even in the form of a chain of tokens. In other words, although their initial correct responses may seem to reflect high reasoning capability, those responses can unravel when subjected to a more rigorous, dynamic dialectical procedure. That is why we need a more structured way to assess the reasoning capability of these models.

## A.5 Correlation Analysis of Dialectical Patterns for MMLU sub-exams

To capture the full spectrum of dependencies in dialectical reasoning performance, including non-linear and non-monotonic relationships, we employ distance correlation analysis Székely et al. (2007). Distance correlation provides a dependency measure with the fundamental property that $\text{dCor}(X, Y) = 0$ if and only if $X$ and $Y$ are statistically independent. $\text{dCor}(X, Y)$ is computed

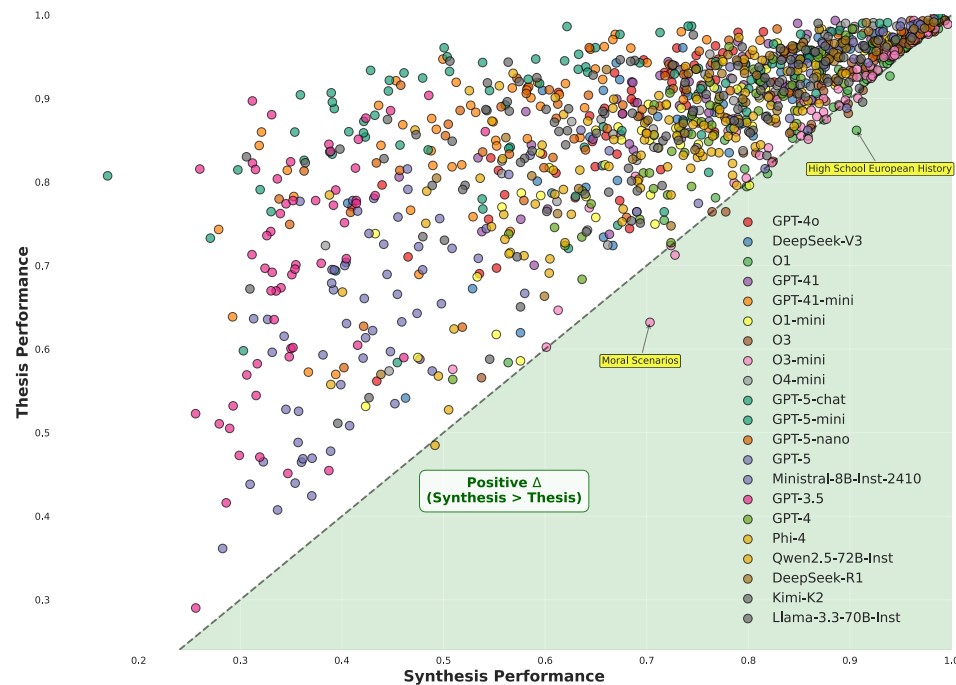

Figure 10: Detailed results of different models in different MMLU exams. Each circle shows the overall result of a model in a sub-topic in MMLU benchmark.

through the following steps:

$$\text{dCor}^2(X, Y) = \frac{\text{dCov}^2(X, Y)}{\sqrt{\text{dVar}(X)\,\text{dVar}(Y)}}, \quad \text{where } \text{dCov}^2(X, Y) = \frac{1}{n^2} \sum_{k,l=1}^{n} A_{kl} B_{kl}.$$

where $A_{kl}$ and $B_{kl}$ are double-centered Euclidean distance matrices Székely & Rizzo (2009).

In what follows we present the distance correlation analysis in MMLU benchmark and its sub-exams. IN particular, we present resutls for (1) overall LLMs: Figure 11, (2) two of top performing models, O1 and GPT-5 (Figure 12 and Figure 13, respectively), (3) two of the middle performing ones, DeepSeek-R1 and Qwen2.5-72B-Instruct (Figure 14 and Figure 15, respectively) (4) two of the low performing ones, GPT-3.5, GPT-5-chat (Figure 16 and Figure 17, respectively).

**General Notes about the Figures**:

- Matrix Structure: Figures show a symmetric correlation matrix displaying all pairwise relationships between four key variables: OC (Oppositional Compliance), Thesis Performance, Synthesis Performance, and $\Delta$ (dialectical gap).

- Diagonal Elements: Histograms show the distribution of each variable, with bins displaying frequency patterns.

- Contour Visualization: Two overlays per scatter plot: (1) Purple/pink confidence ellipses at 50%, 75%, and 95% dependence levels, with darker shades indicating stronger confidence regions; (2) Orange/red density contour lines labeled 25%, 50%, 75%, 90% representing data concentration percentiles, from dark red (highest density core) to light orange (outer data boundary).

- Distance Correlation Values: Each scatter plot displays dCor values (0-1 scale) in the title, capturing both linear and non-linear dependencies. Contour shapes reveal dependency patterns beyond simple linear correlations.

- Symmetry: The matrix exploits correlation symmetry—relationships below the diagonal mirror those above, providing pairwise analysis.

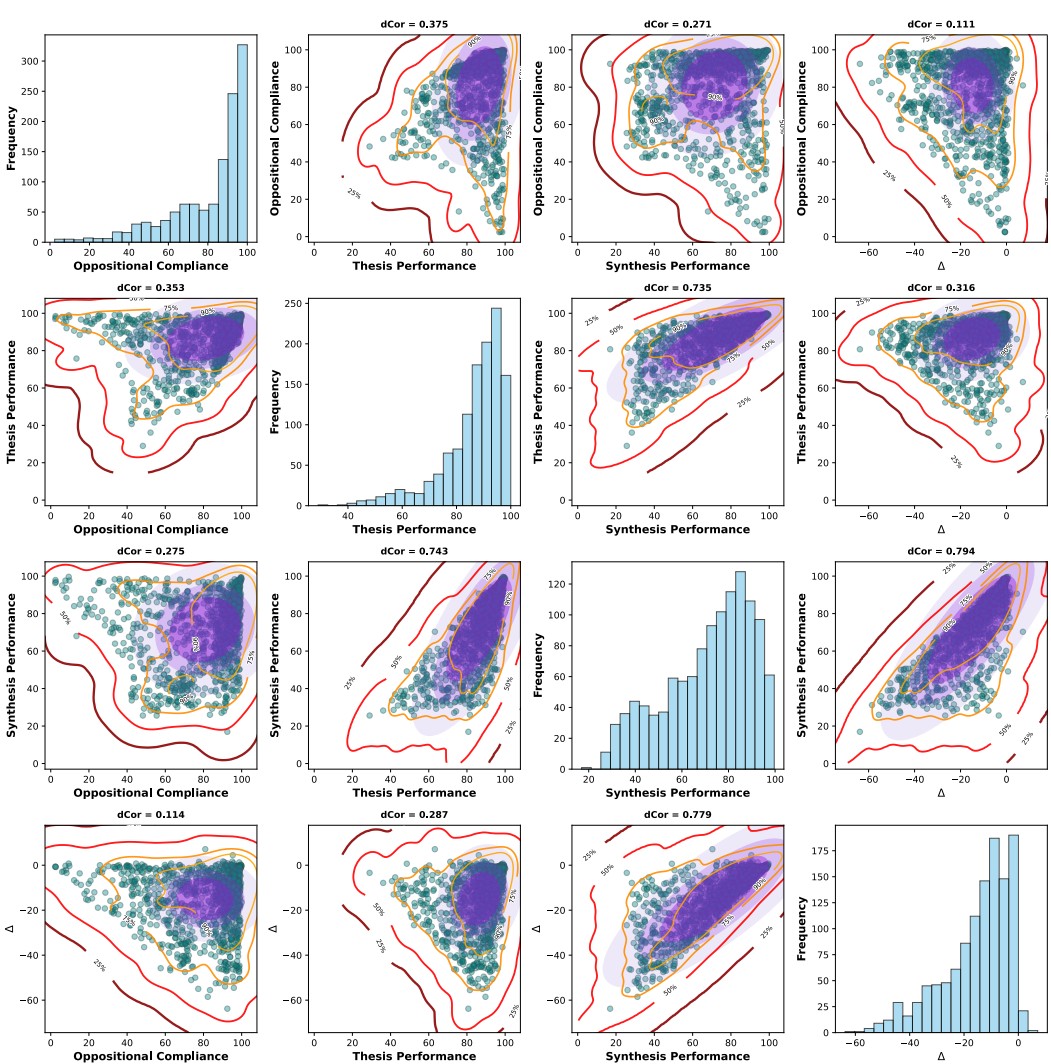

Figure 11: Distance Correlation Analysis of Dialectical Reasoning Performance for all tested LLMs. Distance correlation analysis across MMLU sub-topics reveals weak links between thesis performance and both $\Delta$ and OC, indicating that initial accuracy does not predict dialectical reasoning. As discussed in Section 4.1, low OC often leads to low $\Delta$, suggesting minimal deviation from the thesis when antitheses are weak. While thesis–synthesis correlation is stronger, it remains inconsistent—models with similar thesis scores can yield widely varying synthesis outcomes. Taken together with Section 4.1, these patterns suggest that high thesis accuracy, the conventional way to assess the reasoning capability of LLMs, does not by itself demonstrate genuine reasoning. However, low thesis score reliably flags weak reasoning performance.

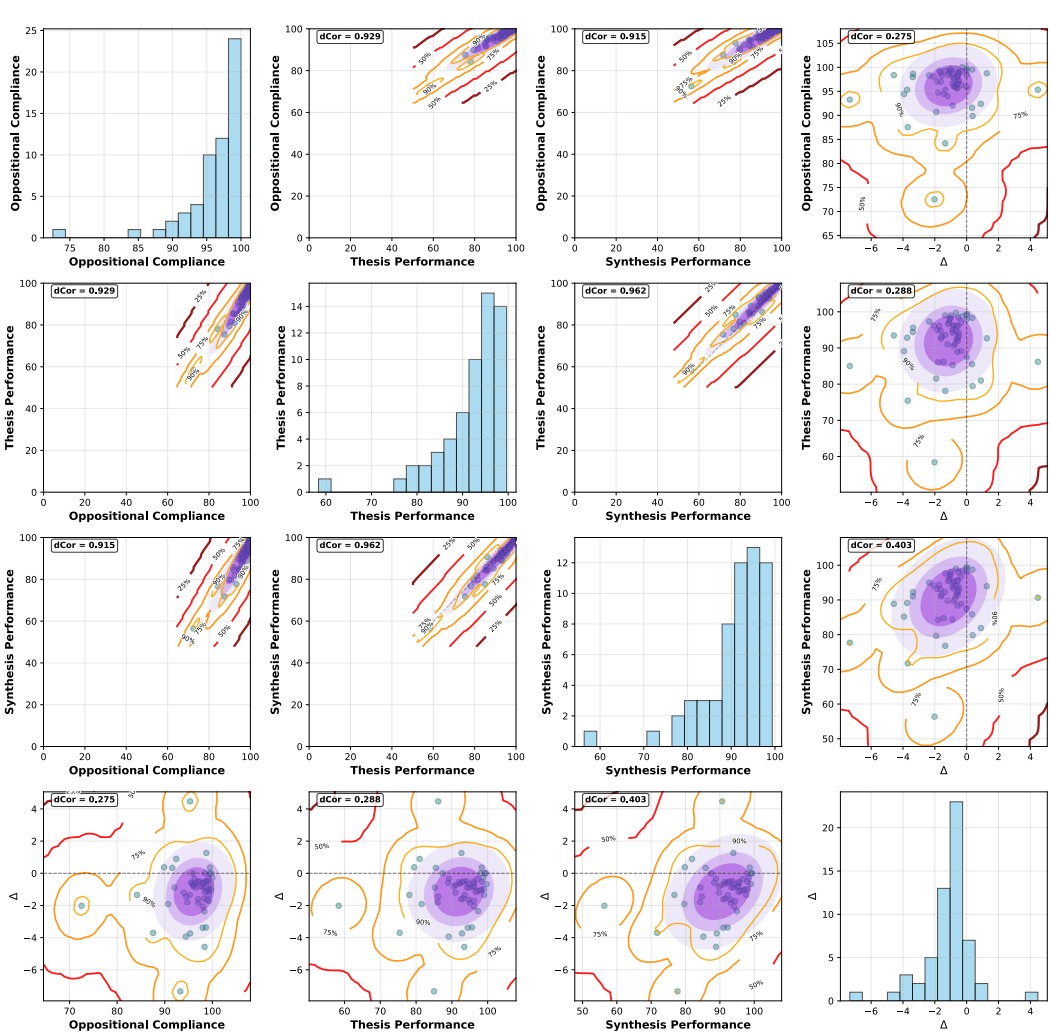

Figure 12: Distance Correlation Analysis of Dialectical Reasoning Performance for O1. The correlation values reveal notable patterns for the O1 model, a top performer. Thesis, OC, and synthesis scores exhibit strong, nearly linear correlations. However, the thesis–$\Delta$ subplot shows that in most cases, the model produces negative $\Delta$ values. This means that although O1 can generate antitheses that effectively challenge the thesis, it often fails to leverage these opposing views to reach a higher-order synthesis—indicating limited dialectical reasoning capability. Moreover, the narrow range of $\Delta$ values suggests that O1 tends to stay close to its original thesis, which explains the strong correlation between thesis and synthesis scores.

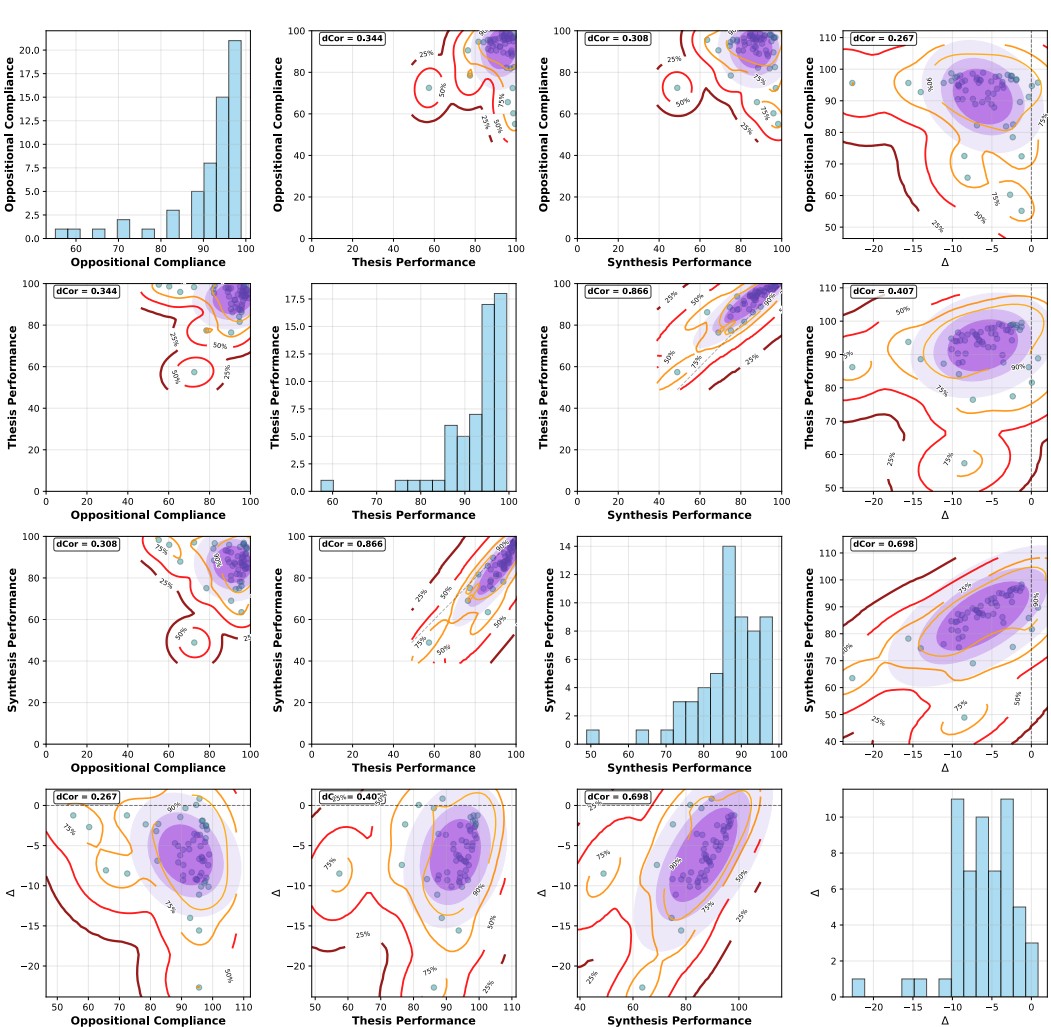

Figure 13: Distance Correlation Analysis of Dialectical Reasoning Performance for GPT-5. The correlation matrix reveals several notable patterns in GPT-5's dialectical reasoning behavior. While the model shows a strong correlation between thesis and synthesis performance, and a moderate link between synthesis and $\Delta$, its ability to perform dialectical reasoning and leveraging oppositional views remains limited. The thesis–$\Delta$ correlation suggests that initial accuracy has some influence on dialectical improvement, but not reliably. More importantly, the weak correlations involving OC vs. $\Delta$ and OC vs. synthesis indicate that GPT-5 struggles to transform strong antitheses into meaningful synthesis. In other words, while the model can generate counterpoints, it often fails to use them constructively to reach higher-order reasoning. This behavior reflects a pattern where GPT-5's synthesis tends to align closely with its thesis, rather than evolving through dialectical engagement. The model's reasoning trajectory appears more static than dynamic, reinforcing the need to evaluate models not just by outcome accuracy, but by their ability to reason through structured and dynamic process.

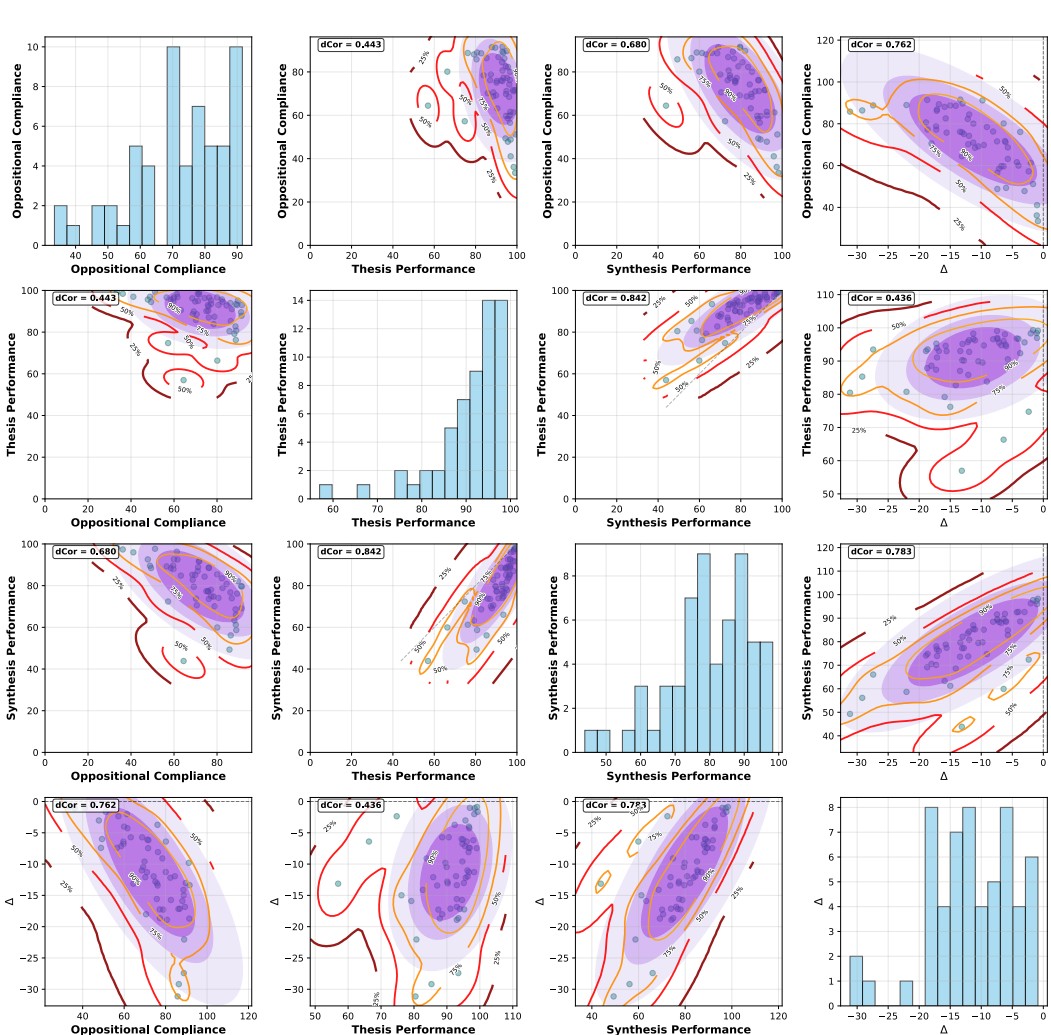

Figure 14: Distance Correlation Analysis of Dialectical Reasoning Performance for DeepSeek R1. The correlation matrix for DeepSeek R1 reveals a more moderate dialectical reasoning profile compared to top-performing models like O1 and GPT-5. In our broader evaluation (Section 4.1), DeepSeek R1 ranks as a middle-tier model, with both its synthesis and OC scores noticeably lower than those of O1 and GPT-5. This is also reflected in the histogram plots: while the thesis scores peak near the maximum, the synthesis scores shift downward, clustering around the 80-point mark. Although DeepSeek R1 shows moderate correlations between thesis, synthesis, and $\Delta$, its ability to make use of opposing views is limited. Similar to the general trend, the weak correlation between thesis and OC suggests that the presence of meaningful opposition in antitheses is not strongly related to the quality of the initial thesis. More importantly, the model often fails to integrate these opposing views into a higher-level synthesis. In many cases, the synthesis either closely mirrors the thesis or drops in quality—sometimes by as much as 30/100 points—highlighting the model's limited capacity for dynamic, dialectical reasoning. Overall, DeepSeek R1's performance reinforces the importance of evaluating models not just by their initial accuracy, but by how effectively they engage with and evolve through structured reasoning processes. Its behavior underscores the need for metrics that go beyond static correctness to assess genuine reasoning capability.

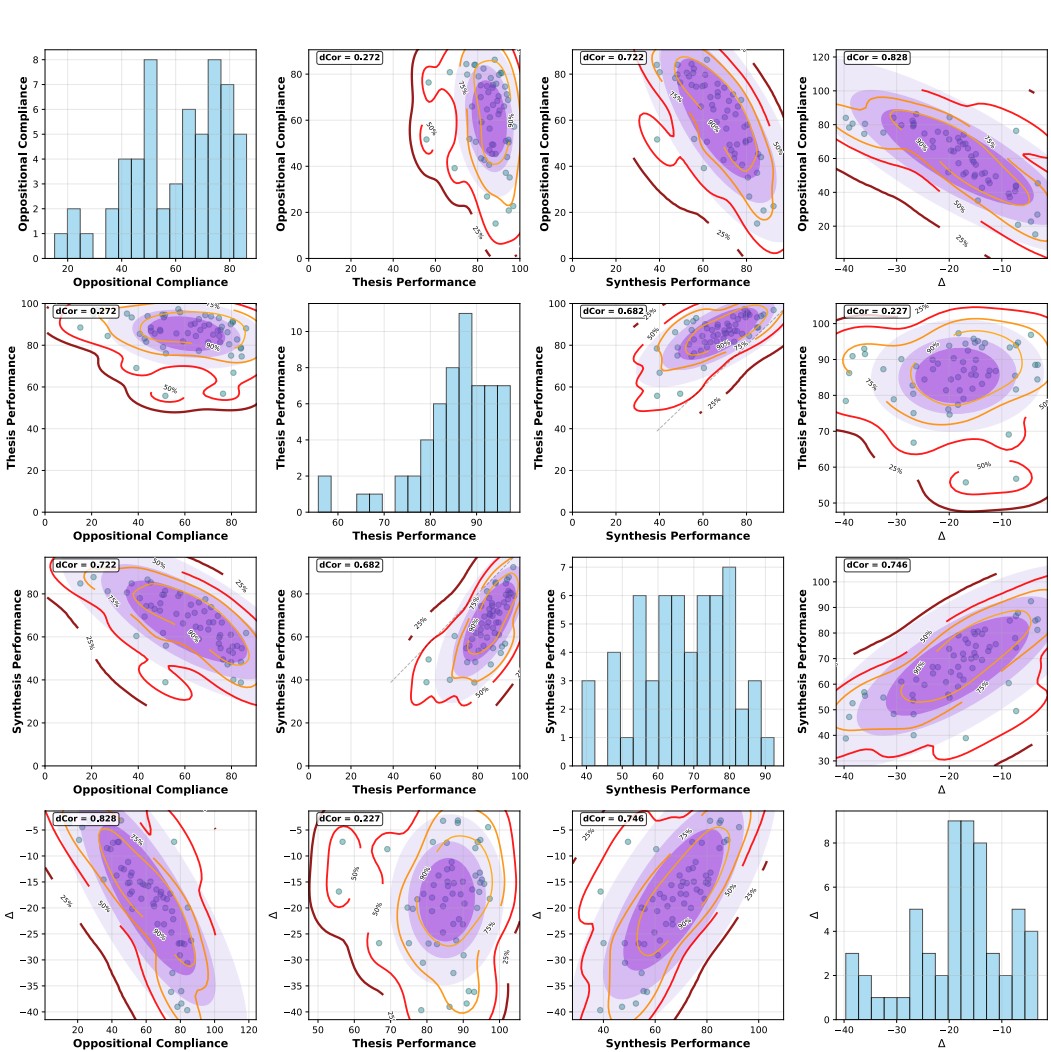

Figure 15: Distance Correlation Analysis of Dialectical Reasoning Performance for Qwen2.5-70B-Instruct. The correlation matrix for Qwen2.5-70B-Instruct reveals a notably weaker dialectical reasoning profile, placing it below DeepSeek R1 in our broader evaluation. Both its synthesis and Oppositional Compliance (OC) scores are lower, as confirmed by the histogram plots: synthesis scores are widely dispersed and skewed toward lower values, while OC—measuring how often antitheses oppose the thesis—is also low and unevenly distributed. The overall correlation pattern resembles that of a mid-performing model like R1, but with more extreme values. For instance, the model's reasoning performance can degrade by as much as 40 points (out of 100), and its OC scores can drop below 20—indicating failure to generate meaningful opposition. Overall, Qwen2.5-70B-Instruct demonstrates how even large-scale models struggle with reasoning when perceived through a dynamic dialectical setting, reinforcing the need for evaluation frameworks that go beyond static accuracy.

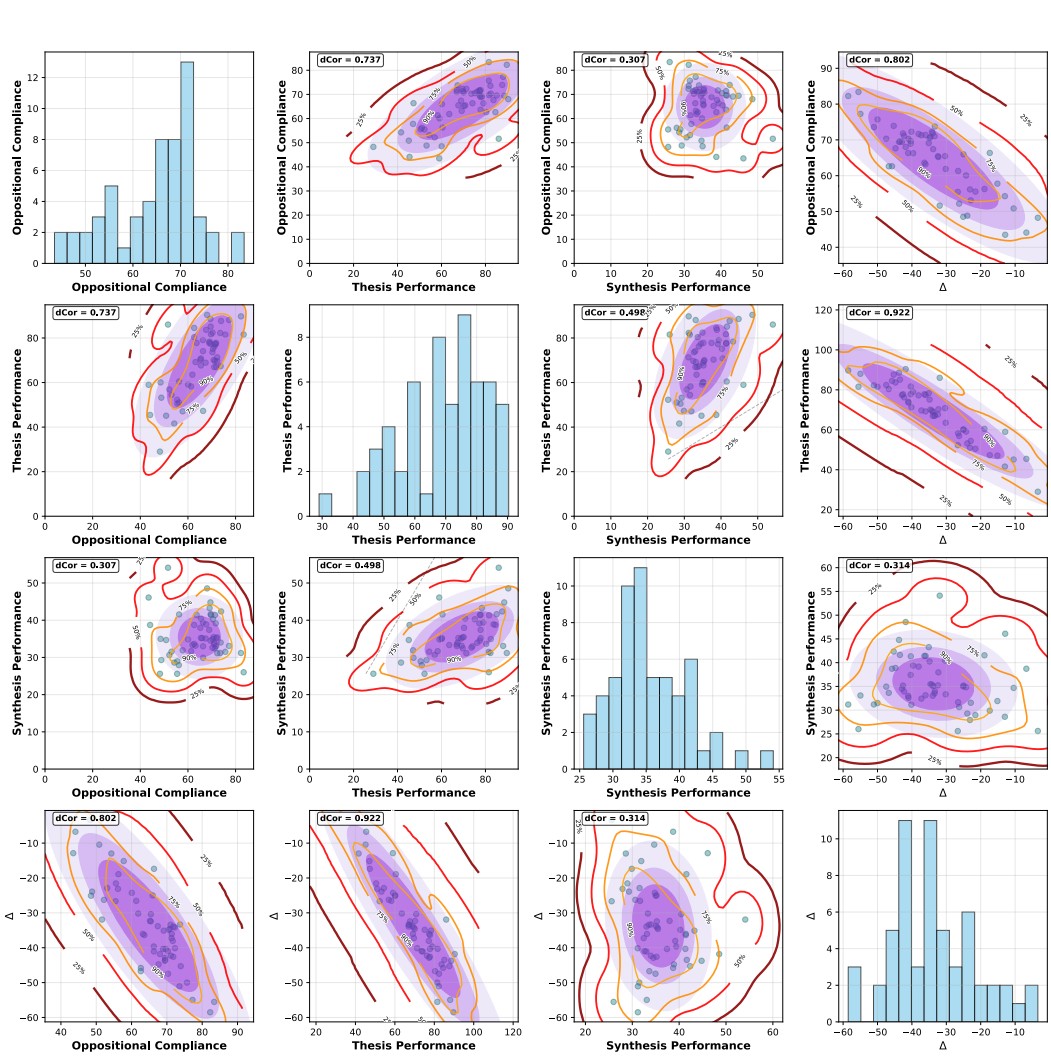

Figure 16: Distance Correlation Analysis of Dialectical Reasoning Performance for GPT-3.5. The correlation matrix for GPT-3.5 reveals the weakest dialectical reasoning profiles among all evaluated models. Across the board, its synthesis and $\Delta$ scores are low, and the histogram plots confirm this: synthesis scores are broadly scattered and skewed toward the lower end. One immediate different pattern comapred to other models is the strong correlation between thesis and $\Delta$. The higher the thesis performance is the lower the $\Delta$ will be. The thesis-OC graph can shed light on this. With increased thesis performance, GPT-3.5 shows increased OC, and consequently with a model that lacks deep reasoning capabilities, it ends up to a lower quality synthesis. As graphs show, the model's performance can degrade as much as 60/100 points in some scenarios. Overall, GPT-3.5 demonstrates limited capacity for dynamic reasoning.

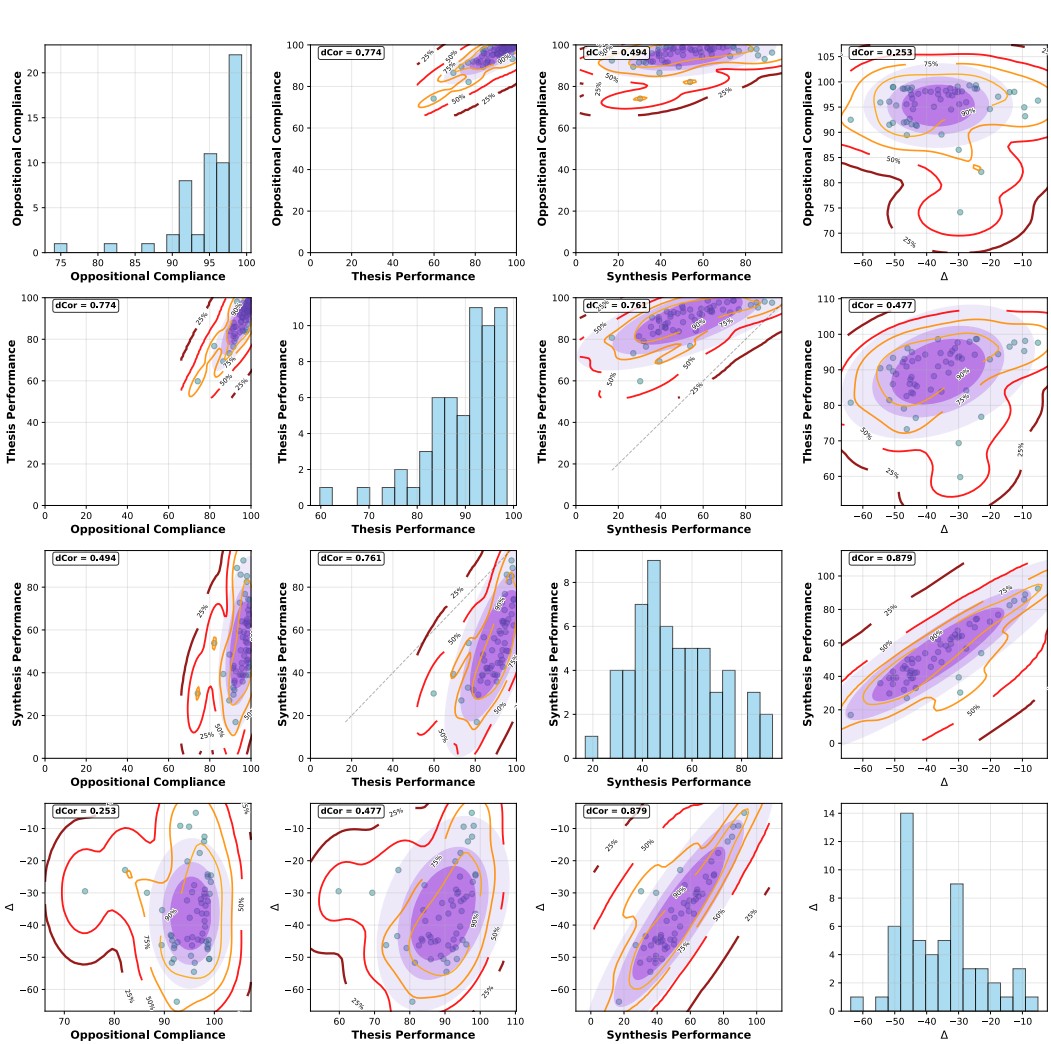

Figure 17: Distance Correlation Analysis of Dialectical Reasoning Performance for GPT-5-chat. The correlation matrix for GPT-5-chat reveals a weak dialectical reasoning profile, placing it among the lowest-performing models in our broader evaluation. Both synthesis and $\Delta$ scores are low, as confirmed by the histogram plots: synthesis scores are broadly scattered and skewed toward the lower range, while $\Delta$ values show limited improvement and frequent degradation. When opposition is present, the model often fails to integrate it into a higher-order synthesis, resulting in reasoning trajectories that either stagnate or degrade (more than 60/100 points degradations in some cases). A notable contrast emerges when comparing GPT-5-chat to GPT-3.5, another low-performing model. GPT-3.5 exhibits a strong correlation between thesis and $\Delta$, where higher thesis scores often lead to greater synthesis degradation due to increased OC. In contrast, GPT-5-chat does not show a strong thesis–$\Delta$ correlation, yet it does exhibit a high correlation between thesis and OC. This suggests that while GPT-5-chat generates more opposition as thesis performance increases, its reasoning trajectory falls apart ending up to have very low synthesis scores. Overall, GPT-5-chat demonstrates limited capacity for dynamic reasoning, yet again, reinforcing the need for evaluation frameworks that go beyond static accuracy and assess how models engage with and evolve through structured dynamic settings.