# OpenReview forum: "Measuring Reasoning in LLMs: a New Dialectical Angle"
_ICLR.cc/2026/Conference — ICLR 2026 Conference Withdrawn Submission_

### Official Review · Reviewer_kPN7 · 2025-10-30

**Soundness:** 3
**Presentation:** 2
**Contribution:** 2
**Rating:** 4
**Confidence:** 4

**Summary:**

This paper proposes SIEV, a framework for evaluating reasoning in large language models (LLMs) through a dialectical process inspired by Hegelian philosophy. Rather than measuring only the correctness of final answers, SIEV introduces a thesis–antithesis–synthesis evaluation procedure, in which models first generate an initial answer (thesis), then an opposing one (antithesis), and finally a reconciled synthesis. The framework is applied to GSM and MMLU benchmarks to assess the "reasoning robustness" of 20+ models, showing that even state-of-the-art models (e.g., GPT-5-chat, DeepSeek-R1) experience large performance drops under this dialectical evaluation.

**Strengths:**

1. **Meaningful topic**: The work tackles an important question: how to evaluate the reasoning process rather than only outcome correctness, which is highly relevant to current debates about reasoning in LLMs.

2. **Interesting idea**: The philosophical grounding in dialectics is creative, offering a novel interpretive angle to assess reasoning dynamics.

3. **Comprehensive experiments**: The evaluation covers a wide range of models and datasets, and the authors provide detailed correlation analyses and visualizations.

**Weaknesses:**

1. **Conceptual mismatch between dialectics and reasoning tasks.**
The dialectical setup is suitable when multiple perspectives coexist (e.g., in debates or moral dilemmas). However, for mathematical reasoning, the solution is typically unique rather than argumentative. In such cases, introducing an artificial "antithesis" is not meaningful: it does not simulate a genuine reasoning conflict but rather fabricates opposition. As shown in Figure 3, the antithesis response often undermines the correct thesis or steers the model toward another incorrect answer. If the thesis is already correct, this step can only destabilize it; if it is wrong, the antithesis merely retrieves a different candidate answer. Consequently, the dialectical process may not reveal genuine reasoning ability but rather random variability in multi-prompt generation.

2. **Limited theoretical justification.**
The paper draws on philosophical concepts (Hegelian dialectics) but does not convincingly show how these translate into a rigorous or computationally meaningful measure of reasoning (some intuitive prompt design is not that convincing). The link between "dialectical synthesis" and improved logical reasoning remains mostly metaphorical.

3. **Evaluation interpretation issues.**
Although the empirical results are extensive, it remains unclear whether the observed "drops" in dialectical scores actually indicate weaker reasoning or simply reflect the instability of multi-turn prompting. The $\Delta$ metric (thesis–synthesis difference) is negative for all models, which may reflect prompt design rather than reasoning limitation.

4. **Scope and motivation.**
The paper’s motivation for using dialectics should be better justified within the context of reasoning evaluation. A more suitable use case might involve open-ended, multi-perspective questions (e.g., debates, ethical reasoning, policy analysis) rather than problems with single correct answers. Repositioning SIEV toward these domains would make the contribution more coherent.

5. **Writing and clarity.**
The paper is generally readable, but long philosophical digressions (Section 3.1) reduce focus. Some expressions are verbose, and minor issues in formatting and grammar could be improved (e.g., line 39).

**Questions:**

See weakness

---

### Official Review · Reviewer_QJTv · 2025-10-31

**Soundness:** 3
**Presentation:** 3
**Contribution:** 3
**Rating:** 4
**Confidence:** 3

**Summary:**

This paper proposes SIEV, a dialectics-based framework for evaluating reasoning in large language models. Instead of only evaluating final answers, SIEV prompts the model to produce a thesis–antithesis–synthesis reasoning pipeline. The method is benchmark-agnostic and applied to GSM and MMLU, revealing large drops in model performance when synthesizing contradictory viewpoints, suggesting current LLMs do not reliably integrate conflicting information. The authors also design metrics such as Opposition Compliance, Dialectic Score, and Δ to quantify dialectical reasoning. Experiments across >20 models show that SIEV uncovers hidden weaknesses in reasoning even in saturated benchmarks and demonstrates cross-model dialectic interaction behaviors.

**Strengths:**

1. Conceptual originality — introduces dialectical reasoning as a structured evaluation lens.
2. Benchmark-agnostic, practical design — works on GSM/MMLU without rewriting tasks, lowering adoption barrier.
3. Reveals hidden weaknesses in models that appear saturated under standard metrics.

**Weaknesses:**

1. Prompt-format sensitivity is not systematically analyzed — reasoning style can heavily depend on prompting, which may affect results.
2. Lack of human or expert annotation to validate that syntheses labeled “failures” truly reflect weak reasoning versus stylistic divergence.
3. The paper largely evaluates structured debate ability, which is one facet of reasoning; broader reasoning types (causal, spatial, inductive) are not covered.

**Questions:**

1. How robust is SIEV to prompt variations or decoding hyperparameters? Have you tried adversarial prompt adjustments to test stability?
2. Is there a calibration mechanism to distinguish true synthesis from verbose compromise or rhetorical averaging?
3. Could human evaluators assess synthesis quality for a subset of results to validate that evaluation metrics align with human judgment?
4. Since Δ < 0 across models, does this reflect model failure or metric design? Why not normalize Δ by thesis difficulty or include reward for holding correct thesis?
5. How might SIEV extend to implicit reasoning tasks (e.g., chain-of-thought hidden) or multi-agent RL settings?

---

### Official Review · Reviewer_DQve · 2025-10-31

**Soundness:** 2
**Presentation:** 2
**Contribution:** 2
**Rating:** 2
**Confidence:** 4

**Summary:**

This paper proposes SIEV, a framework for measuring the reasoning ability of large language models. In SIEV, a model is prompted not only to produce an initial reasoning and answer (thesis), but also to generate an opposing argument (antithesis) and a reconciled conclusion (synthesis) that integrates the two.

**Strengths:**

The idea is intuitively appealing. Treating an LLM like a human reasoner, it makes sense to ask it to defend or refine its position through self-critique.

**Weaknesses:**

1. When the thesis is already correct, forcing the model to generate an antithesis does not seem reasonable. A valid antithesis should not be required to produce a different answer or to "oppose the entirety" of the thesis. Introducing a “forced disagreement” may artificially lower performance.
2. In fact, one could argue that a strong reasoner should maintain its correct stance when challenged, rather than revising it merely for the sake of contradiction. Therefore, a more meaningful experiment would be to challenge the model with an incorrect rationale or wrong answer and evaluate whether it resists adopting the incorrect view, or conversely, whether it can correct a wrong answer when provided with a correct rationale.
3. The negative correlation between $p_T$ and $\delta$ may simply indicate that generating an antithesis is counter-productive, especially given that current LLMs already perform strongly on these datasets.
4. A more faithful metric for measuring reasoning ability would consider only the cases where the thesis is wrong and the synthesis is correct, rather than averaging over all changes.

**Questions:**

How are the prompts for thesis, antithesis, and synthesis decided? Have you tried alternative phrasings or prompt templates, and if so, how much does the performance vary?

---

### Official Review · Reviewer_UUtS · 2025-11-01

**Soundness:** 2
**Presentation:** 2
**Contribution:** 2
**Rating:** 2
**Confidence:** 3

**Summary:**

This paper presents a reasoning process evaluation framework called SIEV. SIEV assesses not only the conclusion a model reaches, but how it gets there. SIEV is built on a well-established philosophical tradition, i.e., dialectics. SIEV is implemented through prompt engineering. Extensive experiments are conducted to evaluate the effectiveness of SIEV.

**Strengths:**

This paper assesses the reasoning process of LLMs from a new angle, i.e., dialectics.

A prompt engineering based framework is proposed to quantify the dialectic score of LLM reasoning.

**Weaknesses:**

This paper is poorly written. Writing style is wordy. A lot paragraphs is spent to elaborate the concept of dialectics. For example, Section 3.1 should state the precisely problem formulation, the metric, etc., instead of state the background.

The technical novelty of this paper is limited. The core part is prompt engineering. The critical flaw is that the quality control of the generated content is not properly treated. In particular, it lacks a mechanism to guarantee the generated content satisfy the desired property of applying dialectics. This makes the evaluation result elusive.

**Questions:**

Please refer to weakness.

---

### Note · Authors · 2025-11-26

**Comment:**

We have decided to withdraw our submission as the review process did not provide proper constructive or technically informed feedback to support a fair evaluation of our work.

**Withdrawal Confirmation:**

I have read and agree with the venue's withdrawal policy on behalf of myself and my co-authors.